# DialoGraph: Incorporating Interpretable Strategy-Graph Networks into Negotiation Dialogues

**Rishabh Joshi, Vidhisha Balachandran, Shikhar Vashishth, Alan W Black, Yulia Tsvetkov**
Language Technologies Institute
Carnegie Mellon University
{rjoshi2, vbalacha, svashish, awb, ytsvetko}@cs.cmu.edu

## Abstract

To successfully negotiate a deal, it is not enough to communicate fluently: pragmatic planning of persuasive negotiation strategies is essential. While modern dialogue agents excel at generating fluent sentences, they still lack pragmatic grounding and cannot reason strategically. We present DialoGraph, a negotiation system that incorporates pragmatic strategies in a negotiation dialogue using graph neural networks. DialoGraph explicitly incorporates dependencies between sequences of strategies to enable improved and interpretable prediction of next optimal strategies, given the dialogue context. Our graph-based method outperforms prior state-of-the-art negotiation models both in the accuracy of strategy/dialogue act prediction and in the quality of downstream dialogue response generation. We qualitatively show further benefits of learned strategy-graphs in providing explicit associations between effective negotiation strategies over the course of the dialogue, leading to interpretable and strategic dialogues.[1]

## 1 Introduction

Negotiation is ubiquitous in human interaction, from e-commerce to the multi-billion dollar sales of companies. Learning how to negotiate effectively involves deep pragmatic understanding and planning the dialogue strategically (Thompson; Bazerman et al., 2000b; Pruitt, 2013).

Modern dialogue systems for collaborative tasks such as restaurant or flight reservations have made considerable progress by modeling the dialogue history and structure explicitly using the semantic content, like slot-value pairs (Larionov et al., 2018; Young, 2006), or implicitly with encoder-decoder architectures (Sordoni et al., 2015; Li et al., 2016). In such tasks, users communicate explicit intentions, enabling systems to map the utterances into specific intent slots (Li et al., 2020). However, such mapping is less clear in complex non-collaborative tasks like *negotiation* (He et al., 2018) and *persuasion* (Wang et al., 2019), where user intent and most effective strategies are hidden. Hence, along with the generated dialogue, the strategic choice of framing and the sequence of chosen strategies play a vital role, as depicted in Figure 1. Indeed, prior work on negotiation dialogues has primarily focused on optimizing dialogue strategies—from high-level task-specific strategies (Lewis et al., 2017), to more specific task execution planning (He et al., 2018), to fine-grained planning of linguistic outputs given

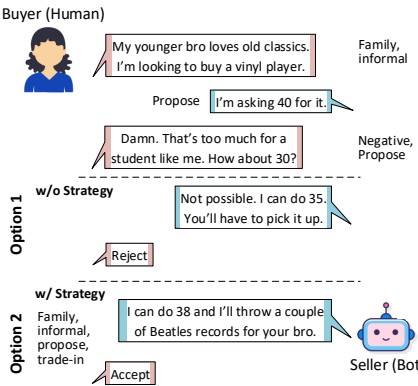

Figure 1: Both options are equally plausible and fluent, but a response with effective pragmatic strategies leads to a better deal.

---

[1]Code, data and a demo system is released at https://github.com/rishabhjoshi/DialoGraph_ICLR21

strategic choices (Zhou et al., 2019). These studies have confirmed that it is crucial to control for pragmatics of the dialogue to build effective negotiation systems.

To model the explicit dialogue structure, prior work incorporated Hidden Markov Models (HMMs) (Zhai & Williams, 2014; Ritter et al., 2010), Finite State Transducers (FSTs) (Zhou et al., 2020) and RNNs (He et al., 2018; Shi et al., 2019). While RNN-based models lack interpretability, HMM- and FST-based approaches may lack expressivity. In this paper, we hypothesize that Graph Neural Networks (GNNs) (Wu et al., 2020) can combine the benefits of interpretability and expressivity because of their effectiveness in encoding graph-structured data through message propagation. While being sufficiently expressive to model graph structures, GNNs also provide a natural means for interpretation via intermediate states (Xie & Lu, 2019; Pope et al., 2019).

We propose DIALOGRAPH, an end-to-end negotiation dialogue system that leverages Graph Attention Networks (GAT) (Veličković et al., 2018) to model complex negotiation strategies while providing interpretability for the model via intermediate structures. DIALOGRAPH incorporates the recently proposed hierarchical graph pooling based approaches (Ranjan et al., 2020) to learn the associations between negotiation strategies, including conceptual and linguistic strategies and dialogue acts, and their relative importance in predicting the best sequence. We focus on buyer–seller negotiations in which two individuals negotiate on the price of an item through a chat interface, and we model the seller's behavior on the CraigslistBargain dataset (He et al., 2018).[2] We demonstrate that DIALOGRAPH outperforms previous state-of-art methods on strategy prediction and downstream dialogue responses. This paper makes several contributions. First, we introduce a novel approach to model negotiation strategies and their dependencies as graph structures, via GNNs. Second, we incorporate these learned graphs into an end-to-end negotiation dialogue system and demonstrate that it consistently improves future-strategy prediction and downstream dialogue generation, leading to better negotiation deals (sale prices). Finally, we demonstrate how to interpret intermediate structures and learned sequences of strategies, opening-up the black-box of end-to-end strategic dialogue systems.

## 2 DIALOGRAPH

We introduce DIALOGRAPH, a modular end-to-end dialogue system, that incorporates GATs with hierarchical pooling to learn pragmatic dialogue strategies jointly with the dialogue history. DIALOGRAPH is based on a hierarchical encoder-decoder model and consists of three main components: (1) *hierarchical dialogue encoder*, which learns a representation for each utterance and encodes its local context; (2) *structure encoder* for encoding sequences of negotiation strategies and dialogue acts; and (3) *utterance decoder*, which finally generates the output utterance. Formally, our dialogue input consists of a sequence of tuples, $\mathcal{D} = [(u_1, da_1, ST_1), (u_2, da_2, ST_2), ..., (u_n, da_n, ST_n)]$ where $u_i$ is the utterance, $da_i$ is the coarse dialogue act and $ST_i = \{st_{i,1}, st_{i,2}, \dots, st_{i,k}\}$ is the set of $k$ fine-grained negotiation strategies for the utterance $u_i$.[3] The dialogue context forms the input to (1) and the previous dialogue acts and negotiation strategies form the input to (2). The overall architecture is shown in Figure 2. In what follows, we describe DIALOGRAPH in detail.

### 2.1 HIERARCHICAL DIALOGUE ENCODER

A dialogue context typically comprises of multiple dialogue utterances which are sequential in nature. We use hierarchical encoders for modeling such sequential dialogue contexts (Jiao et al., 2019). To encode the utterance $u_t$ at time $t$, we use the pooled representations from BERT (Devlin et al., 2019) to obtain the corresponding utterance embedding $e_t$. We then pass the utterance embeddings through a GRU to obtain the dialogue context encoding till time $t$, denoted by $h_t^U$.

---

[2]We focus on the seller's side following Zhou et al. (2019) who devised a set of strategies specific to maximizing the seller's success. Our proposed methodology, however, is general.

[3]For example, in an utterance *Morning! My bro destroyed my old kit and I'm looking for a new pair for $10*, the coarse dialogue act is *Introduction*, and the finer grained negotiation strategies include *Proposing price*, *Being informal* and *Talking about family for building rapport*.

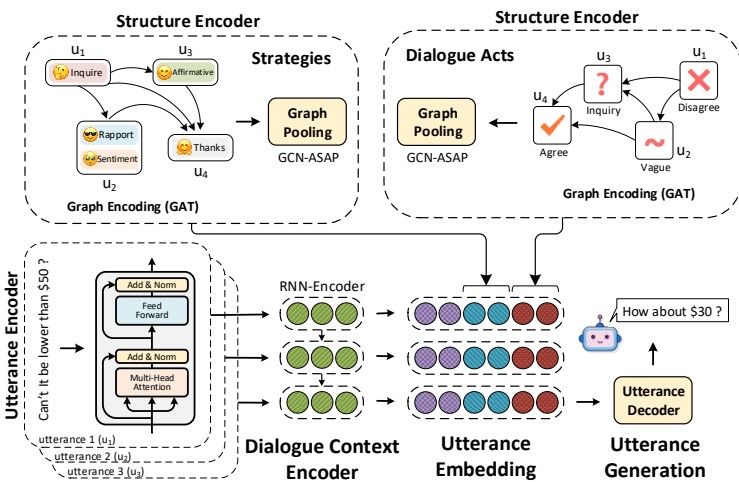

Figure 2: Overview of DIALOGRAPH. At time $t$, utterance $u_t$ is encoded using BERT and then passed to the Dialogue Context Encoder to generate the dialogue representation. This representation is enriched with the encodings of explicit strategy and dialogue act sequences using the structure encoders which is then used to condition the Utterance decoder. Please refer to §2 for details.

## 2.2 STRUCTURE ENCODER

Our structure encoder is designed to model the graph representations of the strategies and dialogue acts using GATs and output their structural representations. These structural representations are used to predict the next set of strategies and dialogue acts and enrich the encoded dialogue representation. Below we describe the structure encoder for negotiation strategies.

We model the sequence of negotiation strategies, $ST = [ST_1, ST_2, \ldots, ST_t]$ by creating a directed graph, where $ST_i$ is the set of k fine-grained negotiation strategies for the utterance $u_i$. Formally, we define a graph $\mathcal{G}(\mathcal{V}, \mathcal{E}, X)$ with $|\mathcal{E}|$ edges and $N = |\mathcal{V}|$ nodes where each node $v_i \in \mathcal{V}$ represents a particular negotiation strategy for an utterance and has a $d$-dimensional feature representation denoted by $\boldsymbol{z}_i$. $\boldsymbol{Z} \in \mathbb{R}^{N \times d}$ denotes the feature matrix of the nodes and $\boldsymbol{A} \in \mathbb{R}^{N \times N}$ represents the adjacency matrix, where $N$ is the total number of nodes (strategies) that have occurred in the conversation till that point. Therefore, each node represents a strategy-utterance pair.

We define the set of edges as $\mathcal{E} = \{(a, b)\}; a, b \in \mathcal{V}$ where $a$ and $b$ denote strategies at utterances $u_a$ and $u_b$, present at turns $t_a$ and $t_b$, such that $t_b > t_a$. In other words, we make a directed edge from a particular node (strategy in an utterance) to all the consecutive nodes. This ensures a direct connection from all the previous strategies to the more recent ones.[4] In the same way, we form the graph out of the sequence of dialogue acts. These direct edges and learned edge attention weights help us interpret the dependence and influence of strategies on each other.

To get the structural representations from the strategy graphs, we pass them through a hierarchical graph pooling based encoder, which consists of $l$ layers of GAT, each followed by the Adaptive Structure Aware Pooling (ASAP) layer (Ranjan et al., 2020). As part of the ASAP layer, the model first runs GAT over the input graph representations to obtain structurally informed representations of the nodes. Then a cluster assignment step is performed which generates a cluster assignment matrix, $S$, which tells the model which nodes come in a similar structural context. After that, the clusters are ranked and then the graph is pooled by taking the top few clusters as new nodes and forming edges between them using the existing graph. This way the size of the graph is reduced at every step which leads to a structurally informed graph representation. We take advantage of the cluster formulation to obtain the associations between the negotiation strategies, as identified from the cluster assignment matrix, $S$. These association scores can later be used to interpret which strategies are associated with each other and tend to co-occur in similar contexts. Moreover, we also use the node attention scores from GAT to interpret the influence of different strategies on the

---

[4]Appendix C shows an example of the graph obtained from a sequence of strategies.

representation of a particular strategy, which essentially gives the dependence information between strategies.

In this way, the structure representation is learned and accumulated in a manner that preserves the structural information (Ying et al., 2018; Lee et al., 2019). After each pooling step, the graph representation is summarized using the concatenation of *mean* and *max* of the node representations. The summaries are then added and passed through fully connected layers to obtain the final structural representation of the strategies $h_t^{ST}$. We employ a similar structure encoder to encode the graph obtained from the sequence of dialogue acts, to obtain $h_t^{da}$.

## 2.3 UTTERANCE DECODER

The utterance decoder uses the dialogue context representation and structural representations of dialogue acts and negotiation strategies to produce the dialogue response (next utterance). We enrich the dialogue representation by concatenating the structural representations before passing it to a standard greedy GRU (Cho et al., 2014) decoder. This architecture follows Zhou et al. (2020), who introduced a dynamic negotiation system that incorporates negotiation strategies and dialogue acts via FSTs. We thus follow their utterance decoder architecture to enable direct baseline comparison. For the $j^{th}$ word of utterance $u_{t+1}$, $w_{t+1}^j$, we condition on the previous word $w_{t+1}^{j-1}$ to calculate the probability distribution over the vocabulary as $p_{t+1}^{w_j} = \text{softmax}(\text{GRU}(h_t, w_{t+1}^{j-1}))$ where $h_t = [h_t^u; h_t^{ST}; h_t^{da}]$ and $[;]$ represents the concatenation operator. For encoding the price, we replace all price information in the dataset with placeholders representing the percentage of the offer price. For example, we would replace \$35 with $< price - 0.875 >$ if the original selling price is \$40. The decoder generates these placeholders which are then replaced with the calculated price before generating the utterance.

## 2.4 MODEL TRAINING

We use $h_t^{ST}$ to predict the next set of strategies $ST_{t+1}$, a binary value vector which represents the k-hot representation of negotiation strategies for the next turn. We compute the probability of the $j^{th}$ strategy occurring in $u_{t+1}$ as $p(st_{t+1,j}|h_t^{ST}) = \sigma(h_t^{ST})$. where $\sigma$ denotes the sigmoid operator. We threshold the probability by 0.5 to obtain the k-hot representation. We denote the weighted negative log likelihood of strategies $\mathcal{L}_{ST}$ as the loss function of the task of next strategy prediction $\mathcal{L}_{ST} = -\sum_j \delta_j \log(p(st_{t+1,j})) - \sum_k \log(1 - p(st_{t+1,k}))$ where the summation of $j$ are over the strategies present ($st'_{t+1,j} = 1$) and not present ($st'_{t+1,k} = 0$) in the ground truth strategies set, $ST'$. Here $\delta_j$ is the positive weight associated with the particular strategy. We add this weight to the positive examples to trade off precision and recall. We put $\delta_j = $ # of instances not having strategy j/# of instances having strategy j.

Similarly, we use $h_t^{da}$ to predict the dialogue act for the next utterance $da_{t+1}$. Given the target dialogue act $da'_{t+1}$ and the class weights $\rho_{da}$ for the dialogue acts, we denote the class-weighted cross entropy loss over the set of possible dialogue acts, $\mathcal{L}_{DA} = -\rho_{da} \log(\text{softmax}(h_t^{da}))$ . We pass $h_t = [h_t^u; h_t^{ST}; h_t^{da}]$ through a linear layer to predict the negotiation success, which is denoted by the sale-to-list ratio $r = (\text{sale price} - \text{buyer target price})/(\text{listed price} - \text{buyer target price})$ (Zhou et al., 2019). We split the ratios into 5 negotiation classes of equal sizes using the training data and use those to predict the success of negotiation. Therefore, given the predicted probabilities for target utterance $u'_{t+1}$ from §2.3, target ratio class $y'_r$ and the learnable parameters $W_r$ and $b_r$, we use the cross entropy loss as the loss for the generation task ($\mathcal{L}_{NLG}$) as well as the negotiation outcome prediction task ($\mathcal{L}_R$), thus $\mathcal{L}_{NLG} = -\sum_{w_j \in u'_{t+1}} \log(p_{t+1}^{w_j})$ and $\mathcal{L}_R = -\sum_{r \in [1,5]} y'_r \log(\text{softmax}(W_r h_t + b_r))$. The $\mathcal{L}_R$ loss optimizes for encoding negotiation strategies to enable accurate prediction of negotiation outcome.

We use hyperparameters $\alpha$, $\beta$ and $\gamma$ to optimize the joint loss $\mathcal{L}_{joint}$, of strategy prediction, dialogue act prediction, utterance generation and outcome prediction together, using the Adam optimizer (Kingma & Ba, 2014), to get $\mathcal{L}_{joint} = \mathcal{L}_{NLG} + \alpha \mathcal{L}_{ST} + \beta \mathcal{L}_{DA} + \gamma \mathcal{L}_R$.

## 3 EXPERIMENTAL SETUP

**Dataset:**    We use the CraigslistBargain dataset[5] (He et al., 2018) to evaluate our model. The dataset was created using Amazon Mechanical Turk (AMT) in a negotiation setting where two workers were assigned the roles of buyer and seller respectively and were tasked to negotiate the price of an item on sale.The buyer was additionally given a target price. Both parties were encouraged to reach an agreement while each of the workers tried to get a better deal. We remove all conversations with less than 5 turns. Dataset statistics are listed in Table 11 in the Appendix.

We extract from the dataset the coarse dialogue acts as described by He et al. (2018). This includes a list of 10 *utterance dialogue acts*, e.g., *inform*, *agree*, *counter-price*. We augment this list by 4 *outcome dialogue acts*, namely, ⟨*offer*⟩, ⟨*accept*⟩, ⟨*reject*⟩ and ⟨*quit*⟩, which correspond to the actions taken by the users. Negotiation strategies are extracted from the data following Zhou et al. (2019). These include 21 fine-grained strategies grounded in prior economics/behavioral science research on negotiation (Pruitt, 2013; Bazerman & Neale, 1993; Bazerman et al., 2000a; Fisher et al., 2011; Lax & Sebenius, 2006; Bazerman et al., 2000b), e.g, *negotiate side offers*, *build rapport*, *show dominance*. All dialogue acts and strategies are listed in Appendices A and B.

**Baselines:**    DIALOGRAPH refers to our proposed method. To corroborate the efficacy of DI-ALOGRAPH, we compare it against our implementation of the present state-of-the-art model for the negotiation task: FST-enhanced hierarchical encoder-decoder model (**FeHED**) (Zhou et al., 2020) which utilizes FSTs for encoding sequences of strategies and dialogue acts.[6] We also conduct and ablation study, and evaluate the variants of DIALOGRAPH with different ways of encoding negotiation strategies, namely, **HED**, **HED+RNN**, and **HED+Transformer**. HED completely ignores the strategy and dialogue act information, whereas HED+RNN and HED+Transformer encode them using RNN and Transformers (Vaswani et al., 2017) respectively. While HED+RNN is based on the dialogue manager of He et al. (2018), HED+Transformer has not been proposed earlier for this task. For a fair comparison, we use a pre-trained BERT (Devlin et al., 2019) model as the utterance encoder (§2.1) and a common utterance decoder (§2.4) in all the models, and only vary the structure encoders as described above. The strategies and dialogue acts in RNN and Transformer based encoders are fed as sequence of $k$-hot vectors.

**Evaluation Metrics:**    For evaluating the performance on the next strategy prediction and the next dialogue act prediction task, we report the F1 and ROC AUC scores for all the models. For these metrics, macro scores tell us how well the model performs on less frequent strategies/dialogue acts and the micro performance tells us how good the model performs overall while taking the label imbalance into account. Strategy prediction is a multi-label prediction problem since each utterance can have multiple strategies. For the downstream tasks of utterance generation, we compare the models using BLEU score (Papineni et al., 2002) and BERTScore (Zhang et al., 2020). Finally, we also evaluate on another downstream task of predicting the outcome of negotiation, using the ratio class prediction accuracy (RC-Acc) (1 out of 5 negotiation outcome classes, as described in §2.4). Predicting sale outcome provides better interpretability over the progression of a sale and potentially control to intervene when negotiation has a bad predicted outcome. Additionally, being able to predict the sale outcome with high accuracy shows that the model encodes the sequence of negotiation strategies well.

## 4 RESULTS

We evaluate (1) strategy and dialogue act prediction (intrinsic evaluation), and (2) dialogue generation and negotiation outcome prediction (downstream evaluation). For all metrics, we perform bootstrapped statistical tests (Berg-Kirkpatrick et al., 2012; Koehn, 2004) and we bold the best results for a metric in all tables (several results are in bold if they have statistically insignificant differences).

**Strategy and Dialogue Act Prediction:**    We compare DIALOGRAPH's effectiveness in encoding the explicit sequence of strategies and dialogue acts with the baselines, using the metrics described in §3. Table 1 shows that DIALOGRAPH performs on par with the Transformer based encoder in

---

[5]`https://github.com/stanfordnlp/cocoa/tree/master/craigslistbargain`

[6]We replace the utterance encoder with BERT for fair comparison. This improved slightly the performance of the FeHED model compared to results published in Zhou et al. (2020).

Table 1: Performance of the next strategy and dialogue-act prediction of various models. We report the F1 and ROC AUC scores. Significance tests were performed as described in §4 and the best results (along with all statistically insignificant values) are bolded.

| | Negotiation Strategies | | | | | | Dialogue Acts | | | | |
| | F1 | | | ROC AUC | | | F1 | | | ROC AUC | |
| Model | Macro | Micro | Weighted | Macro | Micro | Weighted | Macro | Micro | Weighed | Macro | Weighed |
|---|---|---|---|---|---|---|---|---|---|---|---|
| FeHED | 17.6 | 25.6 | 36.3 | 55.8 | 61.7 | 54.7 | 20.6 | 37.4 | 30.6 | 76.9 | 79.2 |
| HED+RNN | 23.2 | 26.7 | 42.4 | 65.3 | 65.3 | 60.4 | 33.0 | **46.2** | 42.8 | 83.1 | 84.2 |
| HED+Transformer | **26.3** | 32.1 | **43.3** | **68.2** | 71.8 | **61.8** | 32.5 | 44.6 | 42.0 | **85.6** | 85.1 |
| DIALOGRAPH | **26.1** | **34.1** | **43.5** | **68.1** | **73.0** | **61.8** | **33.4** | 45.8 | **43.7** | **85.6** | **85.4** |

strategy prediction macro scores and outperforms it on other metrics. Moreover, both significantly outperform the FST-based based method, prior state-of-the-art. We hypothesize that lower gains for dialogue acts are due to the limited structural dependencies between them. Conversely, we validate that for negotiation strategies, RNNs are significantly worse than DIALOGRAPH. We also observe that higher macro scores show that DIALOGRAPH and Transformers are able to capture the sequences containing the less frequent strategies/dialogue acts as well. These results supports our hypothesis of the importance to encode the structure in a more expressive model. Moreover, DIALOGRAPH also provides interpretable structures which the other baselines do not. We will discuss these findings in §5.

**Automatic Evaluation on Downstream tasks:** In this section, we analyze the impact of DIALO-GRAPH on the downstream task of Negotiation Dialogue based on the automatic evaluation metrics described in §3. In Table 2, we show that DIALOGRAPH helps improve the generation of dialogue response. Even though DIALOGRAPH attains higher BLEU scores, we note that single-reference BLEU assumes only one possible response while dialogue systems can have multiple possible responses to the same utterance. BERTScore alleviates this problem by scoring semantically similar responses equally high (Zhang et al., 2020). We also find that both Transformer and DIALOGRAPH have a comparable performance for negotiation outcome prediction, which is significantly better than the previously published baselines (FeHED and HED+RNN). A higher performance on this metric demonstrates that our model is able to encode the strategy sequence better and consequently predict the negotiation outcome more accurately. Additionally, ablation results in Table 3 show that both strategy and dialogue act information helps DIALOGRAPH in improving dialogue response. The difference in BERTScore F1 scores in Tables 2 and 3 arises due to different metrics chosen for early stopping. More details in Appendix D.

Although, both HED+Transformer and DIALOGRAPH are based on attention mechanisms, DIALO-GRAPH has the added advantage of having structural attention which helps encode the pragmatic structure of negotiation dialogues which in turn provides an interpretable interface. The components in our graph based encoder such as the GAT and ASAP layer provide strategy influence and cluster association information which is useful to understand and control negotiation systems. This is described in more detail in §5. Though transformers have self attention, the architecture is limited and doesn't model the structure/dependence between strategies providing only limited understanding. Further, our results show that DIALOGRAPH maintains or improves performance over strong models like Transformer and has much more transparent interpretability. We later show that DI-ALOGRAPH performs significantly better than HED+Transformer in human evaluation.

**Human Evaluation:** Since automatic metrics only give us a partial view of the system, we complement our evaluation with detailed human evaluation. For that, we set up DIALOGRAPH and the baselines on Amazon Mechanical Turk (AMT) and asked workers to role-play the buyer and negotiate with a single bot. After their chat is over, we ask them to fill a survey to rate the dialogue on how persuasive (*My task partner was persuasive.*), coherent (*My task partner's responses were on topic and in accordance with the conversation history.*), natural (*My task partner was human-like.*) and understandable (*My task partner perfectly understood what I was typing.*) the bot was [7]. Prior research in entailment has shown that humans tend to get better as they chat (Mizukami et al., 2016; Beňuš et al., 2011) and so we restrict one user to chat with just one of the bots. We further

---

[7]We use the setup of `https://github.com/stanfordnlp/cocoa/`. Screenshots in Appendix H.

Table 2: Downstream evaluation of negotiation dialogue generation and negotiation outcome prediction. The best results (along with all statistically insignificant values to those) are bolded.

| | | Generation | | | Outcome |
| | | | BERTScore | | Prediction |
| Model | BLEU | Precision | Recall | F1 | RC-Acc |
|---|---|---|---|---|---|
| HED | 20.9 | 21.8 | 22.3 | 22.1 | 35.2 |
| FeHED | 23.7 | 27.1 | 26.8 | 27.0 | 42.3 |
| HED+RNN | 22.5 | 22.9 | 22.7 | 22.8 | 47.9 |
| HED+Transformer | **24.4** | 27.4 | **28.1** | 27.7 | **53.7** |
| DIALOGRAPH | **24.7** | **27.8** | **28.3** | **28.1** | **53.1** |

Table 3: DIALOGRAPH ablation analysis. This shows that all the different components provide complementary benefits. We also evaluate without BERT for comparison with previously published works.

| Model | BERT Score F1 |
|---|---|
| DIALOGRAPH | **27.4** |
| w/o Strategy (ST) | 26.8 |
| w/o ST, Dialogue Acts (DA) | 26.3 |
| w/o ST, DA, BERT | 22.7 |

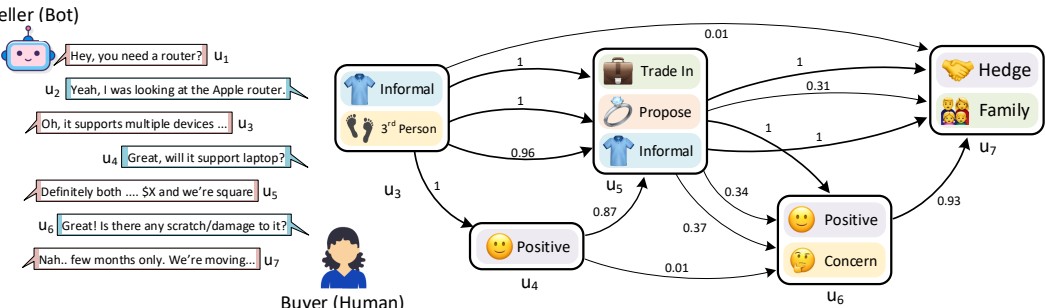

Figure 3: Visualization of the learnt latent strategy sequences in DIALOGRAPH where bolder edges represent higher influence. Here we present only a few edges for brevity and visualize min-max normalized attention values as edge weights to analyze the relative ranking of strategies. For example, for *family* at $u_7$, *informal* of $u_5$ has the most influence followed by *propose*. We present the full attention map for this example in Figure 5 in the Appendix.

prune conversations which were incomplete potentially due to dropped connections. Finally, we manually inspect the conversations extracted from AMT to extract the agreed sale price and remove conversations that were not trying to negotiate at all.

The results of human evaluations of the resulting 90 dialogues (about 20 per model) are presented in Table 4. We find that baselines are more likely to accept unfair offers and apply inappropriate strategies. Additionally, DIALOGRAPH bot attained a significantly higher Sale Price Ratio, which is the outcome of negotiation, showing that effectively modeling strategy sequences leads to more effective negotiation systems. Our model also had a higher average total number of turns and words-per-turn (for just the bots) compared to all baselines, signifying engagement. It was also more persuasive and coherent while being more understandable to the user. From qualitative inspection we observe that the HED model generates utterances that are shorter and less coherent. They are natural responses like "*Yes it is*", but generic and contextually irrelevant. We hypothesize that this is due to the HED model not being optimized to encode the sequence of negotiation strategies and dialogue acts. We believe that this is the reason for the high natural score for HED. From manual inspection we see that HED is not able to produce very persuasive responses. We provide an example of a dialogue in Appendix F. We see that although HED+Transformer model performs well, DIALOGRAPH achieves a better sale price outcome as it tries to repeatedly offer deals to negotiate the price. We see that the HED is unable to understand the user responses well and tends to repeat itself. Both the FeHED and HED baselines tend to agree with the buyer's proposal more readily whereas HED+Transformers and DIALOGRAPH provide counter offers and trade-ins to persuade the user.

## 5 INTERPRETING LEARNED STRATEGY GRAPHS

We visualize the intermediate attention scores generated by the GATs while obtaining the strategy node representations. These attention scores tell us what strategies influenced the representation of a particular strategy and can be used to observe the *dependence* between strategies (cf. Xie & Lu,

Table 4: Human evaluation ratings on a scale of 1-5 for various models. We also provide the average sale price ratio (§2.4). Negative ratio means that average sale price was lower than the buyer's target.

| Model | Persuasive | Coherent | Natural | Understandable | Sale Price Ratio | Avg Turns | Avg words/turn |
|---|---|---|---|---|---|---|---|
| HED | 2.50 | 2.50 | **4.50** | 2.50 | -2.13 | 11.00 | 4.25 |
| FeHED | 3.30 | 3.75 | 3.70 | **3.69** | 0.25 | 14.30 | 5.76 |
| HED+RNN | 2.81 | 3.27 | 3.36 | 3.27 | -3.68 | 13.90 | 3.61 |
| HED+Transformer | 3.50 | 3.50 | 3.70 | 3.40 | -0.07 | 11.40 | 4.36 |
| **DIALOGRAPH** | **3.58** | **3.94** | 3.75 | **3.70** | **0.49** | 15.72 | **5.84** |

Table 5: Examples of strategies and their least / highly associated strategies based on association scores extracted using the cluster attention scores given by the ASAP layer.

| Negotiation Strategy | Least associative strategies | Highly associative strategies |
|---|---|---|
| concern | certainty (0.1759), trade in (0.228) | politeness please (0.7072), politeness gratitude (0.5859) |
| hedge | trade in (0.4367), pos sentiment (0.4501) | propose (0.5427) friend (0.6218) |
| propose | factive count (0.3878), family (0.416) | politeness gratitude (0.5048), trade in (0.5223) |
| negative sentiment | trade in (0.3089), informal (0.3644) | family (0.6363), propose (0.6495) |

2019; Norcliffe-Brown et al., 2018). We show an example in Figure 3 where for brevity, we present a subset of few turns and only the top few most relevant edges in the figure. For visualization, we re-scale the attention values for all incoming edges of a node (strategy) using min-max normalization. This is done because the range of raw attention values would differ based on the number of edges and this allows us to normalize any difference in scales and visualize the relative ranking of strategies (Yi et al., 2005; Chen & Liu, 2004). We notice that as soon as the first *propose* at $u_5$ happens, the strategies completely change and become independent of the strategies before the propose point. From Figure 3, we see that the edge weight from $u_4$ to $u_6$ is 0.01, signifying very low influence. We noticed this trend in other examples as well, wherein, the influence of strategies coming before the first propose turn to strategies coming after that, is very low. A similar phenomenon was also observed by Zhou et al. (2019) who study the conversations by splitting into two parts based on the first propose turn. Another interesting thing we note is that the *trade-in* and *propose* strategies at $u_5$ seem to be heavily influenced by *informal* from $u_3$. Similarly, the *informal* of $u_5$ was influenced by *positive sentiment* from $u_4$. This indicates that the seller was influenced by previous informal interactions to *propose* and *trade-in* at this turn, and that sellers tend to be more informal if the conversation partner is *positive*. In other examples, we see that at a particular utterance, different strategies depend on separate past strategies and also observe that the attention maps usually demonstrate the strategy switch as soon as the first *propose* happens, which is similar to what has been observed by prior work. These examples demonstrate that DIALOGRAPH can model fine-grain strategies, learn dependence beyond just utterances and give interpretable representations, which previous baselines, including the FSTs, lack. Specifically, each state of the FST is explicitly represented by an action distribution which can only be used to see the sequence of strategies and not observe associations or dependence information which DIALOGRAPH provides.

We utilize these cluster attention scores from the ASAP pooling layer to observe the *association* between various strategies which can help us observe strategies with similar contextual behaviour and structural co-occurrence. We take the average normalized value of the cluster attention scores between two strategies to obtain the association score between them. In Table 5, we show some examples of strategies and their obtained association scores. We observe that negative sentiment tends to be most associated to propose. We hypothesize that this is because that people who disagree more tend to get better deals. We observe that people do not tend to associate negative sentiment with trade-in, which is in-fact highly associated with positive sentiment, because people might want to remain positive while offering something. Similarly, people tend to give vague proposals by hedging, for instance, *I could go lower if you can pick it up*, than when suggesting trade-in. Concern also seems to be least associated with certainty, and most with politeness-based strategies. Thus, we observe that our model is able to provide meaningful insights which corroborate prior observations, justifying its ability to learn strategy associations well.

## 6    RELATED WORK

**Dialogue Systems:** Goal-oriented dialogue systems have a long history in the NLP community. Broadly, goal-oriented dialogue can be categorized into *collaborative* and *non-collaborative* systems. The aim of agents in a collaborative setting is to achieve a common goal, such as travel and flight reservation (Wei et al., 2018) and information-seeking (Reddy et al., 2019). Recent years have seen a rise in non-collaborative goal-oriented dialogue systems such as persuasion (Wang et al., 2019; Dutt et al., 2020; 2021), negotiation (He et al., 2018; Lewis et al., 2017) and strategy games (Asher et al., 2016) due to the challenging yet interesting nature of the task. Prior work has also focused on decision-making games such as Settlers of Catan (Cuayáhuitl et al., 2015) which mainly involve decision-making skills rather than communication. Lewis et al. (2017) developed the DealOrNoDeal dataset in which agents had to reach a deal to split a set of items. Extensive work has been done on capturing the explicit semantic history in dialogue systems (Kumar et al., 2020; Vinyals & Le, 2015; Zhang et al., 2018). Recent work has shown the advantage of modeling the dialogue history in the form of belief span (Lei et al., 2018) and state graphs (Bowden et al., 2017). He et al. (2018) proposed a bargaining scenario that can leverage semantic and strategic history. Zhou et al. (2020) used unsupervisedly learned FSTs to learn dialogue structure. This approach, however, although effective in explicitly incorporating pragmatic strategies, does not leverage the expressive power of neural networks. Our model, in contrast, combines the interpretablity of graph-based approaches and the expressively of neural networks, improving the performance and interpretability of negotiation agents.

**Graph Neural Networks:** The effectiveness of GNNs (Bruna et al., 2013; Defferrard et al., 2016; Kipf & Welling, 2017) has been corroborated in several NLP applications (Vashishth et al., 2019), including semantic role labeling (Marcheggiani & Titov, 2017), machine translation (Bastings et al., 2017), relation extraction (Vashishth et al., 2018), and knowledge graph embeddings (Schlichtkrull et al., 2018; Vashishth et al., 2020). Hierarchical graph pooling based structure encoders have been successful in encoding graphical structures (Zhang et al., 2019). We leverage the advances in GNNs and propose to use a graph-based explicit structure encoder to model negotiation strategies. Unlike HMM and FST based encoders, GNN-based encoders can be trained by optimizing the downstream loss and have superior expressive capabilities. Moreover, they provide better interpretability of the model as they can be interpreted based on observed explicit sequences (Tu et al., 2020; Norcliffe-Brown et al., 2018). In dialogue systems, graphs have been used to guide dialogue policy and response selection. However, they have been used to encode external knowledge (Tuan et al., 2019; Zhou et al., 2018) or speaker information (Ghosal et al., 2019), rather than compose dialogue strategies on-the-fly. Other works (Tang et al., 2019; Qin et al., 2020) focused on keyword prediction using RNN-based graphs. Our work is the first to incorporate GATs with hierarchical pooling, learning pragmatic dialogue strategies jointly with the end-to-end dialogue system. Unlike in prior work, our model leverages hybrid end-to-end and modularized architectures (Liang et al., 2020; Parvaneh et al., 2019) and can be plugged as explicit sequence encoder into other models.

## 7    CONCLUSION

We present DIALOGRAPH, a novel modular negotiation dialogue system which models pragmatic negotiation strategies using Graph Attention Networks with hierarchical pooling and learns an explicit strategy graph jointly with the dialogue history. DIALOGRAPH outperforms strong baselines in downstream dialogue generation, while providing the capability to interpret and analyze the intermediate graph structures and the interactions between different strategies contextualized in the dialogue. As future work, we would like to extend our work to discover successful (e.g.: good for the seller) and unsuccessful strategy sequences using our interpretable graph structures.

### ACKNOWLEDGMENTS

The authors are grateful to the anonymous reviewers for their invaluable feedback, and to Alissa Ostapenko, Shruti Rijhwani, Ritam Dutt, and members of the Tsvetshop at CMU for their helpful feedback on this work. The authors would also like to thank Yiheng Zhou for helping with negotiation strategy extraction and FeHED model. This material is based upon work supported by the National Science Foundation under Grant No. IIS2007960 and by the Google faculty research award. We would also like to thank Amazon for providing GPU credits.

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

# A    DIALOGUE ACTS

Here we provide the details about the dialogue acts that we have used to annotate the utterances. 10 are taken from He et al. (2018) and 4 are based on the actions taken by the users. The rule based acts are extracted using the code provided by them[8]. The details are in Table 6.

Table 6: The list of dialogue acts that we use to annotate the data.

| Meaning | Dialogue Act | Example | Detector |
|---|---|---|---|
| Greetings | intro | I would love to buy | rule |
| Ask a question | inquiry | Sure, what's your price | rule |
| Propose the first price | init-price | I'm on a budget so i could do $5 | rule |
| Proposing a counter price | counter-price | How about $15 and I'll waive the deposit | rule |
| Unknown | unknown | Hmm, let me think | rule |
| Agree with the proposal | agree | That works for me | rule |
| Disagree with a proposal | disagree | Sorry I can't agree to that | rule |
| Answer a question | inform | This bike is brand new | rule |
| Using comparatives with existing price | vague-price | That offer is too low | rule |
| Insist on an offer | insist | Still can I buy it for $ 5. I'm on a tight budget | rule |
| Offer the price | ⟨offer⟩ | | agent action |
| Accept the offer | ⟨accept⟩ | | agent action |
| Reject the offer | ⟨reject⟩ | | agent action |
| Quit the session | ⟨quit⟩ | | agent action |

# B    NEGOTIATION STRATEGIES

Here we provide the details about the 15 Negotiation Strategies (Zhou et al., 2019) and 21 Negotiation Strategies (Zhou et al., 2020) in Tables 7 and 8.

Table 7: The details of 15 Negotiation Strategies proposed by Zhou et al. (2019).

| High level Negotiation Rules | Sub Strategy | Example | Detector |
|---|---|---|---|
| Focus on interests, not positions | Describe Product | The car has leather seats | classifier |
| | Rephrase product | 45k miles ⟶ less than 50k miles | classifier |
| | Embellish product | a luxury car with attractive leather seats | classifier |
| | Address concerns | I've just taken it to maintenance | classifier |
| | Communicate interests | I'd like to sell it asap. | classifier |
| Invent options for mutual gain | Propose Price | How about 9k? | classifier |
| | Do not propose first | n/a | rule |
| | Negotiate side offers | I can deliver it for you | rule |
| | Hedge | I **could** come down a bit | rule |
| Build Trust | Communicate Politely | Greetings, gratitude, apology, please | rule |
| | Build rapport | My kid really liked this bike, but he outgrew it | rule |
| | Talk informally | Absolutely, ask away! | rule |
| Insist on your position | Show dominance | The absolute highest I can do is 640 | rule |
| | Negative Sentiment | Sadly, I simply cannot go under 500 | rule |
| | Certainty words | It has **always** had a screen protector | rule |

---

[8] https://github.com/stanfordnlp/cocoa/

| Negotiation Strategies | Train set frequency |
|---|---|
| first_person_singular_count | 26,121 |
| pos_sentiment | 24,862 |
| number_of_diff_dic_pos | 18,610 |
| third_person_singular | 17,000 |
| hedge_count | 12,227 |
| number_of_diff_dic_neg | 10,402 |
| personal_concern | 9,135 |
| propose | 8,449 |
| politeness_greet | 6,639 |
| assertive_count | 4,437 |
| neg_sentiment | 3,680 |
| factive_count | 3,429 |
| politeness_gratitude | 3,171 |
| first_person_plural_count | 2,876 |
| liwc_certainty | 2,530 |
| liwc_informal | 2,396 |
| third_person_plural | 1,721 |
| trade_in | 883 |
| politeness_please | 372 |
| family | 201 |
| friend | 149 |
| <start> | 5,383 |

Table 8: The details of 21 Negotiation Strategies (<start> added by us) used by Zhou et al. (2020). These are used to operationalize the 15 strategies using a rule based system (`https://github.com/zhouyiheng11/augmenting-non-collaborative-dialog/`). The frequency statistics on the train set (5383 conversations) is given. A detailed description regarding the rules used by prior work to extract these are out of scope of this work, however, we intend to provide the code and extracted strategies, along with the rule based mapping to the 15 strategies upon acceptance of this work.

## C  STRATEGY-GRAPH VISUALIZATION

A visualization of a strategy sequence graph. Refer to §2.2 for more details. We also provide additional details regarding the number of nodes and edges in our strategy graphs in Table 9.

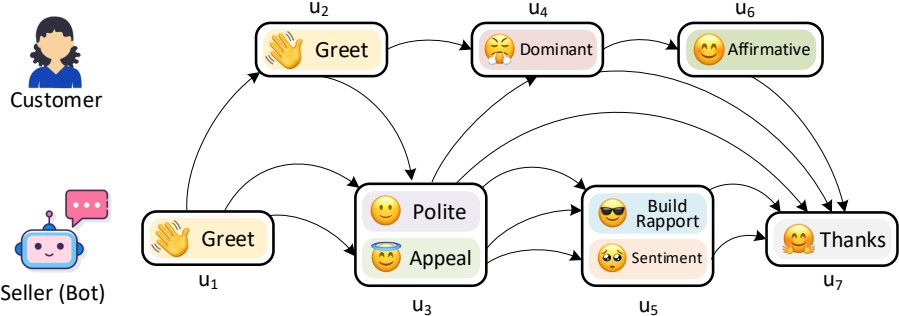

Figure 4: Visualization of a strategy sequence graph. The graph connects each strategy with all previously occurring strategies. Here we present only a few edges for brevity. For example, there would be two more additional edges from $u_4$ to the strategies of $u_5$.

Table 9: We report the number of nodes and edges in our strategy-graphs. Each node corresponds to a particular utterance-strategy pair.

| Feature | Value |
|---|---|
| Max no. of nodes in graph (total strategies) | 86 |
| Avg no. of nodes in graph | 21 |
| Max no. of edges in graph | 3589 |
| Avg no. of edges in graph | 308 |

## D  HYPERPARAMETERS

We present the hyper-parameters for all the experiments, their corresponding search space and their final values in Table 10. We also present additional details of our experiments below. We use most of the hyperparameters from Zhou et al. (2020). Each training run took at most 3 hours on a single Nvidia GeForce GTX 1080Ti GPU and all the models were saved based on Strategy Macro F1 performance.

For experiments for Table 1 and 2 we saved the best models on best Strategy Macro F1 performance (HED being saved on outcome class prediction). This is because we wanted to prioritize and optimize our final model to capture sequence-structural information owing to our focus on interpretability. While performing ablation studies for Table 3, not all models have structure encoders, and hence for a fair comparison we chose a metric independent of the different modules for all the models in ablations. We use the negotiation outcome class prediction (RC-Acc) scores as that optimizes the dialogue for good negotiation outcome, which indirectly helps train the model to capture the sequence of strategies.

Table 10: Here we describe the search-space of all the hyper-parameters used in our experiments.

| Model | Hyper-parameter | Search space | Final Value |
|---|---|---|---|
| All | BERT | - | bert-base-uncased no fine tuning |
| All | BERT Dropout | - | 0.3 |
| All | Dialogue context embedding | - | 300 |
| All | Dialogue context dropout | - | 0.1 |
| All | learning-rate (lr) | 5e-3, 1e-3, 5e-4 | 1e-3 |
| All | max utterances in batch | 64,128,256 | 128 |
| All | weighted strategy loss | True,False | True |
| All | decay rate (l2) | - | 1e-3 |
| All | loss alpha | 1,5 | 1 |
| All | loss beta | - | 10 |
| All | loss gamma | - | 10 |
| All | projection layers for strategy | - | 64 |
| All | projection layers for DA | - | 64 |
| HED+RNN | hidden size | 64, 300 | 64 |
| HED+Transformer | hidden size | 64,300 | 300 |
| HED+Transformer | decoder layers | - | 6 |
| HED+Transformer | attention heads | 1,2 | 2 |
| HED+Transformer | dropout | 0.0, 0.1 | 0.0 |
| DIALOGRAPH | ASAP pooling ratio | 0.2,0.5,0.8 | 0.8 |
| DIALOGRAPH | hidden dim | 64,128 | 64 |
| DIALOGRAPH | Graph layers | 1,2,3 | 2 |
| DIALOGRAPH | Graph dropout | 0.0,0.2 | 0.0 |

# E  NEGOTIATION DATASET STATISTICS

In Table 11 we provide the CraiglistBargain dataset statistics along with data sizes after filtering conversations with less than 5 turns. The maximum and average number of turns in any conversation is 47 and 9.2 respectively. Also, the maximum and average number of strategies in an utterance is 13 and 3 respectively.

Table 11: Dataset statistics.

| Data split | Size |
|---|---|
| Train conversations | 5383 |
| Valid conversations | 643 |
| Test conversations | 656 |
| Filtered train conversations | 4828 |
| Filtered valid conversations | 561 |
| Filtered test conversations | 567 |
| Vocabulary size | 13339 |

# F  EXAMPLE CONVERSATIONS

Table 12: Examples of the generated dialogues of various models when we keep the buyer utterances same. We see that DIALOGRAPH gets the best deal for the same dialogue context and is more persistent. The FeHED and HED models accept offers more readily. We provide more examples of DIALOGRAPH in Table 13.

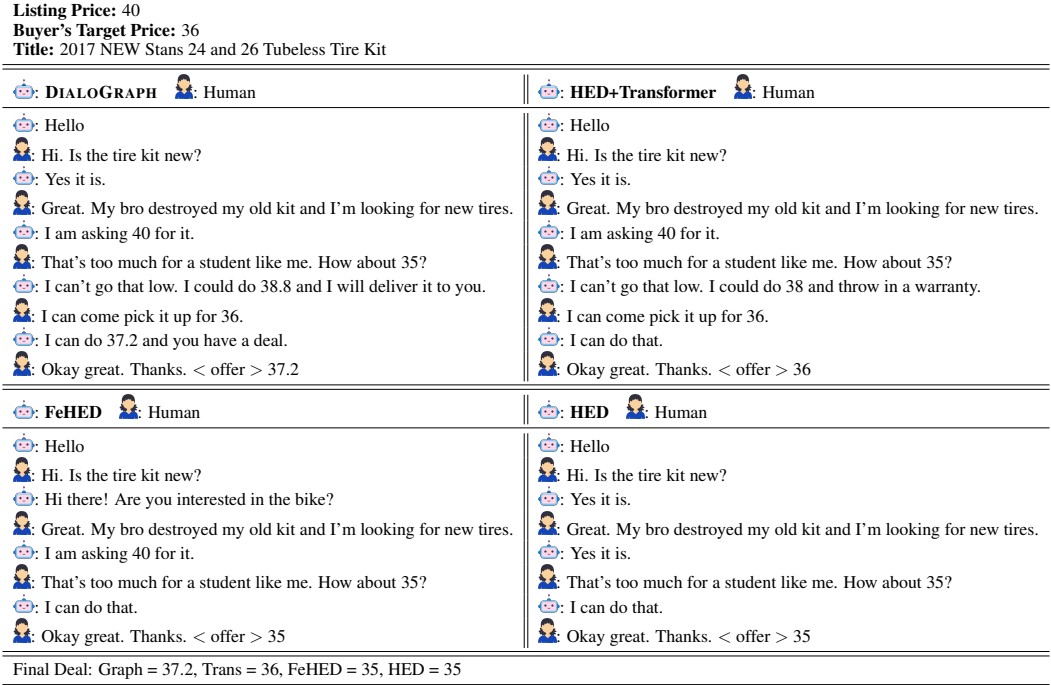

**Listing Price:** 40
**Buyer's Target Price:** 36
**Title:** 2017 NEW Stans 24 and 26 Tubeless Tire Kit

: **DIALOGRAPH**  : Human   ‖ : **HED+Transformer**  : Human

: Hello
: Hi. Is the tire kit new?
: Yes it is.
: Great. My bro destroyed my old kit and I'm looking for new tires.
: I am asking 40 for it.
: That's too much for a student like me. How about 35?
: I can't go that low. I could do 38.8 and I will deliver it to you.
: I can come pick it up for 36.
: I can do 37.2 and you have a deal.
: Okay great. Thanks. < offer > 37.2

: Hello
: Hi. Is the tire kit new?
: Yes it is.
: Great. My bro destroyed my old kit and I'm looking for new tires.
: I am asking 40 for it.
: That's too much for a student like me. How about 35?
: I can't go that low. I could do 38 and throw in a warranty.
: I can come pick it up for 36.
: I can do that.
: Okay great. Thanks. < offer > 36

: **FeHED**  : Human   ‖ : **HED**  : Human

: Hello
: Hi. Is the tire kit new?
: Hi there! Are you interested in the bike?
: Great. My bro destroyed my old kit and I'm looking for new tires.
: I am asking 40 for it.
: That's too much for a student like me. How about 35?
: I can do that.
: Okay great. Thanks. < offer > 35

: Hello
: Hi. Is the tire kit new?
: Yes it is.
: Great. My bro destroyed my old kit and I'm looking for new tires.
: Yes it is.
: That's too much for a student like me. How about 35?
: I can do that.
: Okay great. Thanks. < offer > 35

Final Deal: Graph = 37.2, Trans = 36, FeHED = 35, HED = 35

Table 13: More examples of the generated dialogues of DIALOGRAPH.

| |
|---|
| **Listing Price:** 40 |
| **Buyer's Target Price:** 36 |
| **Title:** 2017 NEW Stans 24 and 26 Tubeless Tire Kit |

🤖 : DIALOGRAPH  👤 : Human

🤖 : Hello
👤 : Hi.
🤖 : Are you interested in the kit?
👤 : Yes I am interested.
🤖 : I am looking for 40 for it.
👤 : No I am willing to pay 35 for it.
🤖 : I can't go that low
👤 : What can you take?
🤖 : I could do 38.8 and I will deliver it to you.
👤 : I only have 36 please.
🤖 : I can do 37.2 and you can pick it up yourself.
👤 : Sorry i only have 36
🤖 : I can do that.
👤 : Okay thank you.
🤖 : < offer > 37.2

🤖 : Hello
👤 : Hi how are you.
🤖 : I am good.
👤 : Is the kit good?
🤖 : yes it is.
👤 : are you sure?
🤖 : yes. i am willing to negotiate the price.
👤 : okay how much you willing to sell.
🤖 : it is in great condition. i am asking 40 for it.
👤 : i will give you 25.
🤖 : i can do 38.8 and you can pick it up.
👤 : 35
🤖 : i can do 37.2 and you can pick it up.
👤 : 36 last
🤖 : ok i can do that.
👤 : < offer > 36

🤖 : Hello
👤 : Hi
🤖 : Hi i see you are interested in my apartment
👤 : i was interested in the tire kit
🤖 : i m asking 40 for it.
👤 : would you be able to do 35?
🤖 : i m willing to negotiate a little bit. what is your budget?
👤 : the highest i can go is 36
🤖 : i cant go that low i could do 38.8
👤 : okay that works
🤖 : < offer > 38.8

## G    INFLUENCE VISUALIZATION

Refer to Figure 5.

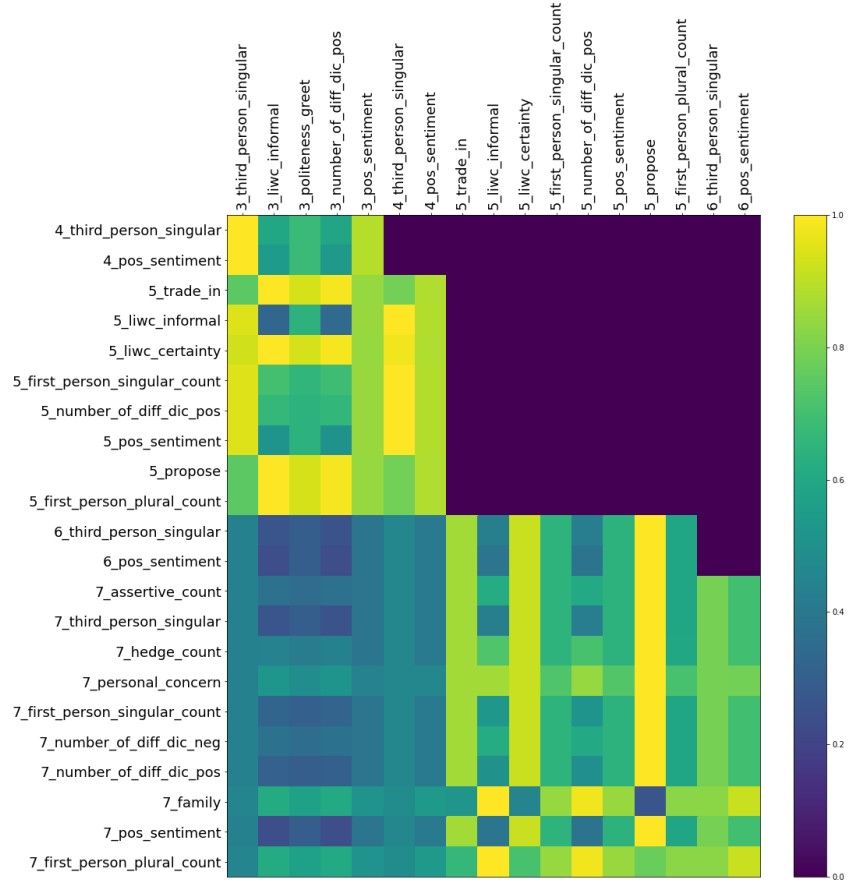

Figure 5: Visualization of the attention map learned by DIALOGRAPH for the example depicted in Figure 3 in the main paper. We only show it for a few turns for brevity. Here the axis labels represent the turn and the strategy. Refer to the Figure 3 in the main paper for description.

## H  HUMAN EVALUATION INTERFACE

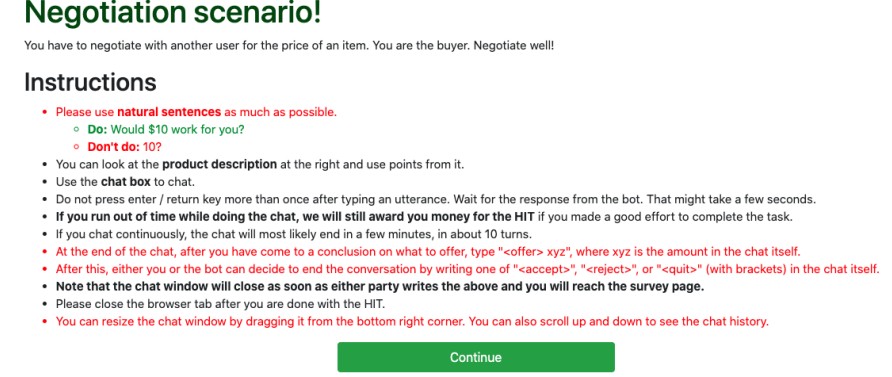

Figure 6: Screenshot of the introduction for the human evaluation interface.

Figure 7: Screenshot of the chat window for the human evaluation interface.

Figure 8: Screenshot of the survey for the human evaluation interface.

