# OpenReview forum: "DialoGraph: Incorporating Interpretable Strategy-Graph Networks into Negotiation Dialogues"
_ICLR.cc/2021/Conference — ICLR 2021 Poster_

### Official Review · AnonReviewer3 · 2020-10-24
**This paper proposes to apply Graph Attention Networks to introduce pragmatic information into negotiation dialogues and achieves good results. The use of Graph networks also makes the relations between strategies more interpretable.**

**Rating:** 5
**Confidence:** 3

**Review:**

* Summary:
This paper proposes to DialoGraph, a model that utilizes Graph Attention Networks to learn the pragmatic relations between strategy and dialogue-act in negotiation dialogues, and achieves better performances than previous baselines. The use of Graph Attention Networks also enables a better interpretation of the relation between strategies.

* Quality:
The motivation to introduce pragmatic information into negotiation dialogues is clear. The model is straightforward and effective. The experiments and analyzes are through. But the idea of directly applying Graph Attention Networks is not super exciting.

* Clarity:
The paper is in general written clearly and easy to follow, with a few points to improve listed below.
1. In 2.4, "$p_j$ is the positive weight associated with the particular strategy", is this $p_j$ the probability of the j-th strategy occurring in $u_{t+1}$ mentioned before?
2. In section 5, "We notice that as soon as the first propose at u5 happens, the strategies completely change and become independent of the strategies before the propose point. "
It's not clear to me why the strategies are independent looking at Figure 3. And also, why is the weight in Figure 3 equal to 1 and the sum of outgoing edges bigger than 1? What's the max and min? And what's the cut-off threshold for the edges in Figure 3?
3. In section 4, "(several results are in bold if they have statistically insignificant differences)"
It's not clear which results are significant and which are not from the tables and the descriptions
4. I'd appreciate more dialogue examples in the appendix.
5. How hard is it to optimize the graph networks? Is the training stable?


* Originality:
The model is not very novel as Graph Networks have been used a lot in dialogue models.


* Significance:
DialoGraph could be potentially useful for other negotiation systems but it requires both strategy and dialogue-act annotations, which may be hard to obtain.

* Pros:
1. The interpretation of the learned strategy graph is useful, but that section needs a bit more work in clarity.
2. The use of Graph Attention Network improves the negotiation dialogue system.

* Cons:
1. The model is applied on one negotiation dataset only and requires both strategy and dialogue-act annotations, which may not be available in other datasets, making the model not so generalizable to other tasks.

2. From the results, it seems the major performance jump comes from using the embeddings from pretrained language models (transformer, BERT, etc)

3. It's not clear which results are significant and which are not.

4. It seems there is no connection between the "strategy structure encoder" and the "dialogue-act structure encoder"? Maybe adding a connection between these two modules would give a better performance since dialogue-act and strategies can be related? Also, are they related at all? If not, why do we need both labels?




* Typo:
missing a period in Human Evaluation "effective negotiation system Our model also..."

---

> ### Author Response · Authors · 2020-11-18
> **Response to AnonReviewer3 - 2**
>
> ====> Regarding significance of the system and use of annotations <====
>
> We acknowledge that in reality the user’s dialogue act and strategies are hidden and not explicitly annotated. [9] and [1] show that explicitly incorporating dialogue acts and negotiation strategies respectively are useful in improving negotiation dialogues. Additionally they propose simple rule based parsers to produce the annotations of the Craigslist Bargain dataset without extensive human labour. We leverage the same data setup and model the sequence of dialogue acts and strategies to improve dialogue generation and end-task negotiation performance. This is very similar to many dialogue works where people make use of annotations for utterances derived from other rule based models ([10]; [11]; [12]). Our work relies on having external DA and strategies information that is easy to obtain using existing parsers but the architecture is not dependent on the specific dialogue acts and strategies used.
>
> ====> Regarding performance from BERT <====
>
> Indeed, BERT gives a huge boost over word embeddings such as GloVe and Word2vec in various applications in NLP and this is the reason for choosing it as our utterance encoder. For fair comparison, we reimplement all the baseline models with BERT as the utterance encoder. We compare them with DialoGraph and see a clear boost in performance (Tables 1, 2 and 4) while using the structural encodings from our graph based approach. In the ablations, we present results without BERT to ground our results with previously published works that did not use BERT. All our results in Tables 1, 2 and 4 have BERT and we show that DialoGraph improves strategy prediction, as well as downstream dialogue performance of generation and negotiation outcome prediction over the chosen baselines. DialoGraph is also significantly better than the baselines in human evaluation. DialoGraph also helps us interpret and learn some of the associations between strategies and observe the influence of previous strategies on predicting future strategies.
>
> ====> Regarding strategy and dialogue act relation <====
>
> Thanks for this suggestion. This is an interesting idea. In the current versions, negotiation strategies can be seen as finer task-specific annotations which are based on negotiation theory, whereas dialogue acts are more coarse dialogue properties. [1] proposed the set of operationalizable negotiation strategies based on theory. As we see from examples in Appendix A and B, dialogue acts are more related to the surface form of an utterance (eg. agree, ask a question) whereas negotiation strategies are more pragmatic (eg. negotiate side offers, show dominance). We show ablation results in Table 3 and show that both are important for good performance. In future, we would like to explore the relationships between the coarse dialogue acts and the finer negotiation strategies.
>
> ====> Regarding typo <====
>
> Thanks for pointing this out. We have corrected it in the updated version.
>
> ====> References <====
>
> [1] Yiheng Zhou, He He, Alan W Black and Yulia Tsvetkov. A dynamic strategy coach for effective negotiation. SigDial 2019.
>
> [2] Taylor Berg-Kirkpatrick, David Burkett, and Dan Klein. An empirical investigation of statistical significance in NLP. EMNLP 2012.
>
> [3] Philipp Koehn. Statistical significance tests for machine translation evaluation. EMNLP 2004.
>
> [4] Yi-Lin Tuan, Yun-Nung Chen, and Hung-yi Lee. DyKgChat: Benchmarking dialogue generation grounding on dynamic knowledge graphs. EMNLP 2019.
>
> [5] Hao Zhou, Tom Young, Minlie Huang, Haizhou Zhao, Jingfang Xu, and Xiaoyan Zhu. Commonsense knowledge aware conversation generation with graph attention. IJCAI 2018.
>
> [6] Deepanway Ghosal, Navonil Majumder, Soujanya Poria, Niyati Chhaya, and Alexander Gelbukh. DialogueGCN: A graph convolutional neural network for emotion recognition in conversation. EMNLP 2019.
>
> [7] Jianheng Tang, Tiancheng Zhao, Chenyan Xiong, Xiaodan Liang, Eric Xing, and Zhiting Hu. Target-guided open-domain conversation. ACL 2019.
>
> [8] Jinghui Qin, Zheng Ye, Jianheng Tang, and Xiaodan Liang. Dynamic knowledge routing network for target-guided open-domain conversation. AAAI 2020.
>
> [9] He He, Derek Chen, Anusha Balakrishnan and Percy Liang. Decoupling strategy and generation in negotiation dialogues. EMNLP 2018.
>
> [10] Abhinav Rastogi, Xiaoxue Zang, Srinivas Sunkara, Raghav Gupta and Pranav Khaitan. Towards Scalable Multi-Domain Conversational Agents: The Schema-Guided Dialogue Dataset. AAAI 2020.
>
> [11] Shachi Paul, Rahul Goel and Dilek Hakkani-Tür. Towards Universal Dialogue Act Tagging for Task-Oriented Dialogues. Interspeech 2019.
>
> [12] Chien-Sheng Wu, Steven Hoi, Richard Socher and Caiming Xiong. TOD-BERT: Pre-trained Natural Language Understanding for Task-Oriented Dialogue. EMNLP 2020.

---

> ### Author Response · Authors · 2020-11-18
> **Response to AnonReviewer3 - 1**
>
> We thank the reviewer for their encouraging words and valuable feedback. We especially would like to thank for the detailed points raised for clarification that have helped us make this manuscript better. We have updated the paper draft to reflect the feedback and we hope that we are able to clarify the questions raised.
>
> ====> Regarding positive weight in 2.4 <====
>
> For the task of next strategy prediction, we use the weighted negative log likelihood or the Binary Cross Entropy loss. We add weights to positive examples to trade of recall and precision for each class (strategy). P_j represents the weight of the positive instance of class j. In our experiments, we put P_j = # of instances not having strategy j / # of instances having strategy j. This is a standard setting for binary cross entropy loss. We have updated the notations and added this clarification in section 2.4 in the updated version.
>
> ====> Regarding independence of strategies from Figure 3 <====
>
> From Figure 3, we see that the edge weight from u_4 to u_6 is 0.01, signifying very low influence. We noticed this trend in other examples as well, wherein, the influence of strategies coming before a “propose point” (the first utterance having a propose strategy) to strategies coming after that, is very low. This trend was also observed by a prior work on Negotiation strategies ([1]), where they observe that humans tend to completely alter their way of talking and the strategies they use after any of the parties proposes the first price for the item. Specifically, people tend to try to build rapport (eg. talk about family) before the propose point and tend to focus on the monetary aspects by using strategies to be more vague or negative after that. We will make this point more clear in the updated version.
>
> ====> Regarding sum of outgoing edges in Figure 3 <====
>
> We visualize the min-max normalized attention values as edge weights and that is why the weights of outgoing edges no longer sums to 1. This is done for better visualization and to normalize between nodes which have a different number of out-going edges. The min and max values lie between 0 and 1. There is no specific cut-off threshold for the edges and we just present some of values in the Figure 3 for brevity. We have included the full attention map for this specific example in the appendix in the updated version.
>
> ====> Regarding bolding and statistical significance in section 4 <====
>
> For all our experiments, we performed the bootstrapped statistical tests ([2]; [3]) and bold the best results for a particular metric. In addition to bolding just the absolute higher numbers, we also bold the numbers that are statistically similar based on our statistical tests. We acknowledge that this was confusing in our previous draft and have made it more clear in the captions in the updated version.
>
> ====> Regarding more dialogue examples <====
>
> We have included more dialogue examples for DialoGraph in the appendix in the updated version. Thanks a lot for the suggestion.
>
> ====> Regarding the optimization of graph networks <====
>
> During our experiments, we observed that all our models converged within 10 epochs on a single Nvidia 1080 ti GPU taking 2-3 hours training time. We ran every model 5 times and observed no instability during the training process with insignificant variance in the final results.
>
> ====> Regarding originality and the use of graphs in dialogue <====
>
> Training to control for the pragmatics of the non-collaborative dialogue systems has been less studied, and it is a crucial and interesting topic to build a logical dialog system. In dialogue systems, graphs have been used to guide dialogue policy and response selection. However, they have been used to encode external knowledge ([4]; [5]) or speaker information [6], rather than compose dialogue strategies on-the-fly. Other works ([7]; [8]) focused on keyword prediction using RNN-based graphs. Our work is the first to incorporate GATs with hierarchical pooling, learning pragmatic dialogue strategies jointly with the end-to-end dialogue system. Therefore, although our work is not the first to use graph networks in dialogue systems, our approach of using graph networks for modeling the sequences of strategies and dialogue acts in a non-collaborative dialogue system is novel. Also, our modeling of graph structure helps us interpret and learn some of the associations between strategies and observe the influence of previous strategies on predicting future strategies, something which has not been studied previously.

---

### Official Review · AnonReviewer4 · 2020-10-24
**DialoGraph: Review**

**Rating:** 6
**Confidence:** 4

**Review:**

This paper deals with the problem of natural language generation for a dialogue system involved in complex communication tasks such as negotiation or persuasion. The proposed architecture consists of two encoders: one for the utterance and the other for dialogue acts and negotiation strategies. The decoder is an RNN that converts the encoded vectors to the output utterance. Each utterance is first passed through BERT to get an utterance-level encoding. The sequence of utterance encodings is then passed through an RNN to generate a conversation level encodings. The negotiation strategies and dialogue acts in a conversation are represented using a node-edge graph, where the nodes are one of the N different strategies/acts and there exists an edge from node a to node b if an utterance with strategy A precedes any utterance with strategy B. The entire architecture is trained in a multi-task setup where the loss function accounts for both the predictions of the model and generated language. The proposed architecture is evaluated on the CraigslistBargain dataset and compared against Zhou et al. 2020.

The paper is very clearly written and the experimental work has sufficient detail to ensure reproducibility. The main contribution and the novelty of this paper is in the use of graph neural networks for encoding dialogue acts and negotiation strategies. This choice was mainly because it helps with better interpretability of predictions and this is demonstrated anecdotally in section 5. The proposed model shows better performance in three different metrics when compared to sota from Zhou et al: (1) prediction of dialogue acts and negotiation strategies, (2) on the downstream task of dialogue generation, and (3) human evaluation to quantify the quality of generated language.

There are a few aspects of the paper unclear to me and could use more insight from the authors. (2) The input to the GNN is a node-edge graph where the edges exist between dialogue acts or negotiation strategies based on their precedence order in the conversation. It would be useful to explain why the authors chose this type of representation. What other types of representations were considered? (2) The authors use two different encoders for dialogue acts and negotiation strategies. Would it make sense to have a graphical representation that captures both dialogue acts and negotiation strategies simultaneously? (3) From my understanding, it seems the dialogue acts were annotated in the original work of (He He 2018) and strategies are obtained based on the models and rules published in (Zhou et al 2019). I am not sure if the model evaluation should entail predicting negotiation strategies which are in itself predictions of a different model.

Minor comments/questions:
1. How were the train/test/dev splits done?
2. If I am not mistaken, it seems like the model uses predictions from time step (t) to predict and generate for time step (t+1). Does this mean errors in one of the earlier timesteps could lead to more errors in subsequent timesteps?
3. I think it would be clearer if you were to use a single-letter variable for strategies. It helps for better readability.
4. References: Please use the peer-reviewed version of the paper as opposed to the arxiv version when available. Eg. He He et al 2018.
5. How many conversations were used in human evaluations?

---

> ### Author Response · Authors · 2020-11-18
> **Response to AnonReviewer4 - 2**
>
> ====> Question regarding error propagation by using predictions of time step t to generate time step t+1 <====
>
> We indeed use the predictions from time step (t) to predict and generate for time step (t+1). This is similar to the standard sequence prediction task setting, for example in language models, where the predictions till time step t are utilized to predict the next word. This is a standard setup for utterance generation in dialogue systems. All these systems are indeed prone to error propagation. This is tackled using teacher forcing (giving the gold output for time step (t) as input to time step (t+1) during training time. We also follow a similar training setup where we perform teacher forcing with randomness (instead of the gold output of time step (t), input the predictions of that time step with a specific probability), which encourages the model to generalize to noisy predictions. During inference time, as the standard setup, we just rely on the predictions of time step t.
>
> ====> Regarding notation of strategies <====
>
> We chose this notation to be consistent with prior works that used “da” for dialogue acts and “st” for strategies. We have updated the notations in section 2 to be more consistent and hope that has cleared the confusions.
>
> ====> Regarding references <====
>
> Thanks for pointing this out. We have updated and verified all the references in the updated version.
>
> ====> Regarding number of conversations used in human evaluation <====
>
> We did the human evaluation on 90 valid dialogues from all models.
>
> ====> References <====
>
> [1] Yiheng Zhou, He He, Alan W Black and Yulia Tsvetkov. A dynamic strategy coach for effective negotiation. SigDial 2019.
>
> [2] He He, Derek Chen, Anusha Balakrishnan and Percy Liang. Decoupling strategy and generation in negotiation dialogues. EMNLP 2018.
>
> [3] Yiheng Zhou, Yulia Tsvetkov, Alan W Black and Zhou Yu. Augmenting non-collaborative dialog systems with explicit semantic and strategic dialog history. ICLR 2020.
>
> [4] Abhinav Rastogi, Xiaoxue Zang, Srinivas Sunkara, Raghav Gupta and Pranav Khaitan. Towards Scalable Multi-Domain Conversational Agents: The Schema-Guided Dialogue Dataset. AAAI 2020.
>
> [5] Su Nam Kim, Lawrence Cavedon and Timothy Baldwin. Classifying Dialogue Acts in One-on-one Live Chats. EMNLP 2010.
>
> [6] Chandrakant Bothe, Cornelius Weber, Sven Magg and Stefan Wermter. EDA: Enriching Emotional Dialogue Acts using an Ensemble of Neural Annotators. LREC 2020.
>
> [7] Jun Deng, Xinzhou Xu, Zixing Zhang, Sascha Frühholz and Björn Schuller. Semisupervised Autoencoders for Speech Emotion Recognition. ACM IEEE Transactions on Audio, Speech, and Language Processing.
>
> [8] Shachi Paul, Rahul Goel and Dilek Hakkani-Tür. Towards Universal Dialogue Act Tagging for Task-Oriented Dialogues. Interspeech 2019.
>
> [9] Chien-Sheng Wu, Steven Hoi, Richard Socher and Caiming Xiong. TOD-BERT: Pre-trained Natural Language Understanding for Task-Oriented Dialogue. EMNLP 2020.

---

> ### Author Response · Authors · 2020-11-18
> **Response to AnonReviewer4 - 1**
>
> We thank the reviewer for their encouraging words and valuable feedback. We are glad that the reviewer found our paper clear and our experiments sufficient. We have updated the paper draft to reflect the feedback and we hope that we are able to clarify the questions raised.
>
> ====> 1) Regarding the graph representations and precedence order <====
>
> Motivated by the fine grained nature of negotiation strategies, where an utterance can have multiple possible strategies, we consider our node representations over individual strategies instead of utterances. Since dialogue has a natural precedence order, and intuitively, the choice of what to say next (or what strategies to use next) depends on how the conversation is going up till that point, we model the graph on the precedence order of strategies and dialogue acts. Such a representation can be useful to understand the dependence of a future strategy on the previously occurring strategies. Works on building recommendation systems to assist negotiation [1] have also made use of the history of strategies to predict the next ones. Further, since dialogue and conversations have a precedence nature, it makes sense to learn how to predict the optimum future strategies to improve negotiation. Because of these reasons, we considered a precedence order based graph representation. We experimented with only connecting strategy-nodes over a window size but observed that having long range connections was much more useful.
>
> ====> 2) Regarding encoding dialogue act and strategies simultaneously <====
>
> In the current versions, negotiation strategies can be seen as finer task-specific annotations which are based on negotiation theory, whereas dialogue acts are more coarse dialogue properties. [1] proposed the set of operationalizable negotiation strategies based on theory. As we see from examples in Appendix A and B, dialogue acts are more related to the surface form of an utterance (eg. agree, ask a question) whereas negotiation strategies are more pragmatic (eg. negotiate side offers, show dominance). We show ablation results in Table 3 and show that both are important for good performance. Thanks for the interesting suggestion. In future, we will explore the relationships between the coarse dialogue acts and the finer negotiation strategies by utilizing a similar architecture to learn a common graphical representation.
>
> ====> 3) Regarding annotations and model evaluation <====
>
> [2] and [1] show that explicitly incorporating dialogue acts and negotiation strategies respectively is useful in improving negotiation dialogues. Additionally they propose simple rule based parsers to produce the annotations of the Craigslist Bargain dataset. We leverage the same data setup as [3] and model the sequence of dialogue acts and strategies to improve dialogue generation and end-task negotiation performance. This is very similar to many dialogue works where people make use of annotations for utterances derived from other rule based or neural models ([4]; [5]; [6]; [7]). Our work relies on having external DA and strategies information that is easy to obtain using existing parsers but the architecture is not dependent on the specific dialogue acts and strategies used. Other works have also utilized such soft-labels to evaluate properties of dialogue systems by also confirming the benefits using other methods like human evaluation ([7]; [8]; [9]). [8] and [9] merge dialogue act labels using intuition and rules between datasets and evaluate further models on the updated labels.
>
> We acknowledge that our results for just strategy and dialogue act prediction (Table 1) would not justify the efficacy of DialoGraph in modeling the structural information, since the labels are not exactly human labeled (though the careful rule based annotation process followed by [1] and [2] can be approximated to human labelling), our results on the downstream generation, negotiation outcome prediction and human evaluation (Table 2 and 4) do support our hypothesis that DialoGraph indeed captures strategy and dialogue act information well. Our interpretation results (Figure 3 and Table 5), which are in accordance with prior work also confirm our hypothesis that such a system could be applied to understand pragmatic structural information of sequences of dialogue acts and strategies.
>
> ====> Question regarding train test split <====
>
> The train/test/dev splits were provided by the original authors [2] of CraigslistBargain Dataset.

---

### Official Review · AnonReviewer1 · 2020-10-27
**dialogue system that leverages Graph Attention Networks to model complex negotiation strategies**

**Rating:** 6
**Confidence:** 4

**Review:**

This paper proposes a end-to-end dialogue system that leverages Graph Attention Networks to model complex negotiation strategies. The main contributions that the author claims are to model negotiation strategies through a GNN and using these learned strategies to predict future strategies and generate a response leads to better negotiations.

The end to end model contains a traditional hierarchical encoder to obtain contextual representations along with a structure encoder that is designed to model strategies and dialog acts and obtain structural representations. The decoder is a simple GRU that produces the response by conditioning it on the contextual and structural representation along with the previous word.

Strength & Weakness:
1.  The aspect of generating a response based on the prediction of negotiation strategies and dialog acts is an interesting approach.
2. The results from Table 1 show that transformers perform comparatively or even better than DialoGRAPH. What is the overall gain on using the GNN if the transformers match their performance?
3. The results in Tables 2 & 3 are hard to interpret. Prior research has shown that BLEU and other automated metrics are not good enough to evaluate the performance of dialog systems. Why is the BERT F1 Score for Dialograph in Table 2 and 3 different? What is the major difference?
4. What were the definitions provided for persuasive, coherence, natural, and understandable? Why is there a huge dropoff on the metric for naturalness from HED to other models? How many participants were recruited?

Questions:
1. What is the rationale behind 5 negotiation classes? Was any ablation done with this to determine the optimal number or how this split affects the outcomes?
2. What is the overall size of the vocabulary during decoding?
3. How were the 4 outcome dialog acts labeled across the dataset? was this done through a human annotation process?
4. What decoding strategy was used?


Suggestion:
1. Section 2.4 has inconsistencies in the usage of notations. Please fix those issues.
2. Can the bolding of scores in Table 1 be made more consistent and highlight only those scores that are higher and lower based on the metrics being represented

=======================After reading the authors response ============================
I thank the authors for answering all the questions that been raised by the fellow reviewers. Looking at the responses and changes made to the paper, I have increased the score from 5 to 6 after the authors clarified the issues I had with the paper. Overall this paper demonstrates the effectiveness of using a GNN for negotiation dialogues. I feel that this approach can be applied for any non-collaborative dialog settings and the claims of interpretability make this approach better.

---

> ### Author Response · Authors · 2020-11-18
> **Response to AnonReviewer1 - 2**
>
> ====> Regarding HED naturalness score and number of participants <=====
>
> From qualitative inspection we observe that the HED model generates utterances that are shorter and less coherent. They are natural responses like “Yes it is”, but generic and contextually irrelevant. We hypothesize that this is due to the HED model not being optimized to encode the sequence of negotiation strategies and dialogue acts. We believe that this is the reason for a high natural score for HED. From manual inspection, we see that HED is not able to produce very persuasive responses and relies on generic responses. For example HED produces responses like : “I can do that.”, “It is new.”, “Thank you.”, “That works for me”, etc. The example in the appendix shows cases where HED uses generic statements. We have added this note in the human evaluation results section of the final version.
> We recruited 90 people to annotate the dialogues. Prior research in entailment has shown that humans tend to get better as they chat ([3]; [4]) and so we restrict one user to chat with just one of the bots. This is to avoid the performance difference caused by exposing the users to the bots in specific orders.
>
> ====> Question regarding rationale behind 5 negotiation classes <====
>
> We did a manual pilot study using 20 dialogues to understand the different kinds of negotiation conversations. While prior work explored just classifying the outcomes as either good or bad (for the seller), we observed that there are many ambiguous cases. To account for such varying levels of negotiation quality, we  decided to bucket the outcomes into 5 splits. This number is a hyper parameter which is dataset dependent and can be altered depending on the kind of dialogue system being developed. Also, thank you for the suggestion, we will explore the impact of varying number of splits and incorporating the outcome information in other ways (possibly optimized using RL) in future work.
>
> ====> Question regarding vocabulary size <====
>
> The vocabulary size was 10339. We have included this in the Table 11 - Dataset statistics in the updated version.
>
> ====> Question regarding outcome dialogue acts <====
>
> In the original data collection setup by [1], at the end of each conversation the buyer and the seller had to select options out of <offer>, <accept>, <reject> and <quit>. This information is available in the dataset and we use it to augment our dialogue acts.
>
> ====> Question regarding decoding strategy <====
>
> We utilize a standard greedy GRU decoder for our utterance decoder similar to our baseline FeHED ensuring a fair comparison. We have added this in section 2.3 in the updated version.
>
> ====> Suggestion regarding notation <====
>
> Thanks for the suggestion. We have updated the notations throughout section 2.
>
> ====> Suggestion regarding bolding and statistical significance <====
>
> For all our experiments, we performed the bootstrapped statistical tests ([5]; [6]) and bold the best results for a particular metric. In addition to bolding just the absolute higher numbers, we also bold the numbers that are statistically similar based on our statistical tests. We acknowledge that this was confusing in our previous draft and have made it more clear in the captions in the updated version.
>
> ====> References <====
>
> [1] He He, Derek Chen, Anusha Balakrishnan and Percy Liang. Decoupling strategy and generation in negotiation dialogues. EMNLP 2018.
>
> [2] Yiheng Zhou, Yulia Tsvetkov, Alan W Black and Zhou Yu. Augmenting non-collaborative dialog systems with explicit semantic and strategic dialog history. ICLR 2020.
>
> [3] Masahiro Mizukami, Koichiro Yoshino, Graham Neubig, David Traum, and Satoshi Nakamura. Analyzing the effect of entrainment on dialogue acts. SigDial 2016.
>
> [4] Štefan Beňuš, Agustín Gravano, and Julia Hirschberg. Pragmatic aspects of temporal accommodation in turn-taking. Journal of Pragmatics. 2017.
>
> [5] Taylor Berg-Kirkpatrick, David Burkett, and Dan Klein. An empirical investigation of statistical significance in NLP. EMNLP 2012.
>
> [6] Philipp Koehn. Statistical significance tests for machine translation evaluation. EMNLP 2004.

---

> > ### Comment · AnonReviewer1 · 2020-11-24
> > **Update**
> >
> > Authors,
> > Thanks for providing detailed responses to my comments.  I read your response regarding the advantage GNN provides over transformers but the results from the paper show that transformers match GNN and also achieves a slightly better performance in terms of predicting the overall outcome of the negotiation process. With regards to your rationale for splitting into 5 buckets, I am not 20 dialogs would provide a large enough sample size to make the judgment in terms of the number of buckets. With regards to Table 4, what is the point for measuring avg turns? Does it indicate that negotiations are completed with 11.4 turns for transformers? It would have been nice if the success rate of these negotiations had been measured.

---

> > > ### Author Response · Authors · 2020-11-25
> > > **Response to the Update of AnonReviewer1**
> > >
> > > We thank the reviewer for the additional comments. Regarding the performance difference between DialoGraph and HED+Transformers, after employing the statistical significance tests, we see that the performance of HED+Transformers and DialoGraph is statistically insignificant (and therefore similar) for the outcome prediction metric. As we see from the other results, DialoGraph is either comparable or better than HED+Transformers in all the metrics we evaluated on, while also being significantly better in human evaluation. GNNs also have the added advantage of providing interpretable representations, so even though some metrics show that Transformers have similar performance (insignificant differences), we believe that our graph based approach holds an edge.
> > >
> > > Regarding the sample size for deciding the outcome split into 5 classes : We model the setting based on observations from the 20 observed dialogues and we will explore the splits further with bigger pilot studies. In our data analysis, we observed that many conversations don’t have very obvious good sellers or buyers (final sale price ratio inclined towards the sellers or buyers - good sellers mean a higher sale price ratio). For this reason we decide to split the conversations into using the sale price ratio into more finer splits, than just 2 (good sellers or buyers) as used by prior work. This is done to capture signals from obvious conversations separately from the more confusing ones. Since this information is just used in the loss function, it essentially acts as an inductive bias in our architecture, to nudge the model into differentiating between the different classes of conversations. We evaluate the accuracy of predicting this to decide if the model is able to sufficiently learn the differences, since different types of conversations can potentially have different types of optimum strategy sequences. We will vary the number of splits of the conversation and add the ablation for the same in the camera ready version.
> > >
> > > We measure average turns to indicate which agent is able to hold conversations longer. Even though shorter conversations are preferred in general in task based dialogue systems, for negotiation bots it's preferred to have longer conversations as they indicate more potential for back and forth discussion over the price of the object between the agent and the buyer. Thus, a model with higher average turns might indicate more attempts to negotiate, something that we also see from the qualitative example presented in the Appendix. This metric was also used to evaluate prior works on negotiation dialogues.
> > >
> > > Regarding success rate : For measuring the success rate (or the negotiation success), we measured the average sale price ratio of the chats that were completed. This tells us how well the agent is able to negotiate with the buyer or essentially how successful is the agent in negotiation. From the results we see that DialoGraph achieves the highest sale price ratio signifying that chats with DialoGraph resulted in better negotiation deals.

---

> ### Author Response · Authors · 2020-11-18
> **Response to AnonReviewer1 - 1**
>
> We thank the reviewer for their encouraging words and valuable feedback. We are glad that the reviewer found our approach interesting. We have updated the paper draft to reflect the feedback and we hope that we are able to clarify the questions raised.
>
> ====> Regarding GNN vs Transformers <====
>
> The advantages of using a GNN based approach over transformers is that they have structural attention which helps encode the pragmatic structure of negotiation dialogues which in turn provides an interpretable interface. The components in our graph based encoder such as the GAT and ASAP layer provide strategy influence (Figure 3) and cluster association (Table 5) information respectively, which is useful to understand the strategies and control negotiation systems. Specifically, these tell what strategies are associated with what other strategies and what strategies influence the prediction of future strategies. Though transformers have self attention, the architecture is limited and doesn’t model the structure/dependence between strategies providing only limited understanding. Further, in Tables 1 and 2, we show that DialoGraph maintains or improves performance over a strong model like Transformer and has much more transparent interpretability. DialoGraph is also significantly better than HED+Transformer in human evaluation (Table 4). We have added this clarification in more detail in the results section.
>
> ====> Regarding results of Table 2 and 3 <=====
>
> Table 2 focuses on the downstream evaluation performance of dialogue generation (BLEU and BertScore) and negotiation outcome prediction. To keep our work more aligned with prior work, we evaluate on BLEU while we acknowledge that a single reference BLEU is not a very good metric to evaluate dialogue utterances. For this reason, we make use of BERTScore, which alleviates the problem of single reference BLEU. Since automatic metrics only give us a partial view of the system, we complement our evaluation with detailed human evaluation in Table 4, which also shows that DialoGraph performs better than other baselines.
> Table 3 has ablation experiments of our DialoGraph setup after removing specific parts of our architecture. This helps us see that the different chosen architecture choices contribute to the final performance and that all the components provide complementary benefits. We describe it in more detail in the next point. We have updated the Automatic Evaluation results section in the updated version to clarify these points.
>
> ====> Regarding difference in BERT F1 Scores <=====
>
> The difference arises due to different metrics chosen for early stopping based on the performance on the validation set. For experiments for Table 1 and 2 we saved the best models on best Strategy Macro F1 performance (HED being saved on outcome class prediction). This is because we wanted to prioritize and optimize our final model to capture sequence-structural information owing to our focus on interpretability.
> While performing ablation studies for Table 3, not all models have structure encoders, and hence for a fair comparison we chose a metric independent of the different modules for all the models in ablations. We use the negotiation outcome class prediction (RC-Acc) scores as that optimizes the dialogue for good negotiation outcome, which indirectly helps train the model to capture the sequence of strategies. We have updated the details in appendix D.
>
> ====> Regarding definitions for survey questions <=====
>
> For the human evaluation, we present the following question surveys at the end of each dialogue to the humans, which they are expected to give a response on a standard 5 point Likert Scale rating.
> Coherence : My task partner’s responses were on topic and in accordance with the conversation history.
> Understandable : My task partner perfectly understood what I was typing.
> Natural : My task partner was human-like.
> Persuasiveness : My task partner was persuasive.
> These were the evaluation criteria presented in [1] and [2] and we leverage their setup (https://github.com/stanfordnlp/cocoa/tree/master/craigslistbargain) for a fair comparison. We apologize for omitting the survey questions and have updated the human evaluation section in our draft with these questions. We have also added screenshots of the human evaluation setup in the appendix.

---

### Official Review · AnonReviewer2 · 2020-10-29
**Well-written paper, Interesting topics and ideas.**

**Rating:** 6
**Confidence:** 5

**Review:**

- Summary
This paper presents DialoGraph, which introduces graph structures to encode the relations of strategies and dialog acts among utterances in a dialog history. This paper empirically demonstrated the efficacy of DialoGraph on non-collaborative negotiation dialog tasks, and the model is evaluated automatically and human-evaluation.

- Strong points
	1. Training to control for the pragmatics of the negotiation dialogues has been less studied, and it is a crucial and interesting topic to build a logical dialog system.
	2. This paper is well structured and well written.
	3. Introducing GAT and ASAP to model strategies and dialog acts is novel, and the interpretation of their results is interesting.
	4. This paper also provides soundly experimental results and comparison with other SOTA models.

- Weak points
	1. The dialog task problem setting seems unrealistic. A tuple of utterance, dialog act, and strategy at turn $I$ is given, and based on the previous tuple sequence the model predicts the next response, dialog act, and strategy. However, in reality, the user’s dialog act and strategy are hidden.
	2. Also, the sets of dialog acts and strategies are different depending on user and system. I agree this paper followed the task as previously defined, but it seems awkward.
	3. HED+Transformer vs. DialoGraph: Those two models show similar experimental results. According to the experiment configuration, the HED+Transformer used 6 decoder layers, whereas the DailoGraph used 2 graph layers. Those two models were fairly comparable with respect to the number of parameters? Or, the proposed method based on GAT+ASAP does not effective as much as Transformer, because both are basically based on attention mechanisms?
	4. In table 4, the average words per turn is required to be separately reported depending on users and bots.

- Questions
	1. How the bot decide the price to offer? Does it solely depend on the language model (i.e., the decoder)?
	2. How the bot encode the listed price?
	3. In table 4, why HED got a remarkably higher score on “natural” measure than other models?
	4. Please address and clarify the weak points above.

---

> ### Author Response · Authors · 2020-11-18
> **Response to AnonReviewer2 - 2**
>
> ====> References <====
>
> [1] He He, Derek Chen, Anusha Balakrishnan and Percy Liang. Decoupling strategy and generation in negotiation dialogues. EMNLP 2018.
>
> [2] Yiheng Zhou, He He, Alan W Black and Yulia Tsvetkov. A dynamic strategy coach for effective negotiation. SigDial 2019.
>
> [3] Abhinav Rastogi, Xiaoxue Zang, Srinivas Sunkara, Raghav Gupta and Pranav Khaitan. Towards Scalable Multi-Domain Conversational Agents: The Schema-Guided Dialogue Dataset. AAAI 2020.
>
> [4] Shachi Paul, Rahul Goel and Dilek Hakkani-Tür. Towards Universal Dialogue Act Tagging for Task-Oriented Dialogues. Interspeech 2019.
>
> [5] Chien-Sheng Wu, Steven Hoi, Richard Socher and Caiming Xiong. TOD-BERT: Pre-trained Natural Language Understanding for Task-Oriented Dialogue. EMNLP 2020.

---

> ### Author Response · Authors · 2020-11-18
> **Response to AnonReviewer2 - 1**
>
> We thank the reviewer for their encouraging words. We are glad that the reviewer felt that our paper was well written and has interesting insights and nice results. We also thank the reviewer for their valuable feedback. We have updated the paper draft to reflect the feedback and we hope that we are able to clarify the questions raised.
>
>
> ====> Regarding the task setting <====
>
> We acknowledge that in reality the user’s dialogue act and strategies are hidden and not explicitly annotated. [1] and [2] show that explicitly incorporating dialogue acts and negotiation strategies respectively are useful in improving negotiation dialogues. Additionally, they propose simple rule based parsers to produce the annotations of the Craigslist Bargain dataset without extensive human labour. We leverage the same data setup and model the sequence of dialogue acts and strategies to improve dialogue generation and end-task negotiation performance. This is very similar to many dialogue works where people make use of annotations for utterances derived from other rule based models ([3]; [4]; [5]). Our work relies on having external DA and strategies information that is easy to obtain using existing parsers but the architecture is not dependent on the specific dialogue acts and strategies used.
>
> ====> Regarding transformer vs DialoGraph <====
>
> Although both the Transformer and DialoGraph are based on attention mechanisms, DialoGraph has the added advantage of having structural attention which helps encode the pragmatic structure of negotiation dialogues which in turn provides an interpretable interface. We leverage GAT to obtain influence scores of predicting future strategies and the ASAP layer to obtain cluster associations (Table 5) between strategies, which can tell us what strategies are associated with what other strategies. Further, in Tables 1 and 2, we show that DialoGraph maintains or improves performance over a strong model like Transformer and has much more transparent interpretability. DialoGraph is also significantly better than HED+Transformer in human evaluation (Table 4).  For HED+Transformer, we picked the best performing model with standard 6 decoding layers which had about 11.4M parameters. We hypothesize that the structural attention and hierarchical pooling makes DialoGraph more expressive due to having more inductive bias in the form of pooling of the graphs into smaller graphs. This way, at every layer, the size of the graph decreases substantially while the hierarchical pooling layer captures the coarse graph structure. Our best performing model has 2 graph pooling layers (10.4M parameters) and adding more layers led to overfitting on the dataset.  We have added more details regarding this point in the results section.
>
> ====> Regarding average words per turn <====
>
> Our motivation to measure the average words per turn was to measure the diversity in response generation by different models. The stats reported in Table 4 only corresponds to the average number of words per turn for the bot. A higher average words per turn number tells us that our DialoGraph model does not resort to simple responses and is able to generate longer and more complex utterances, possibly capturing more negotiation strategies. We have added this clarification to the results section in the updated version.
>
> ====> Question regarding price decoding <====
> Yes, the final price that the bot decides depends on the language model. We utilize a standard GRU decoder where all the price information is replaced with a placeholder which represents the percentage of the offer price for a particular product. For example, we would replace 35 with < price - 0.875 >  if the original selling price is 40. The decoder generates these placeholders which are then replaced with the calculated price before generating the utterance. We have added this clarification in section 2.3 in the updated version.
>
> ====> Question regarding HED naturalness score <====
>
> From qualitative inspection, we observe that the HED model generates utterances that are shorter and less coherent. They are natural responses like “Yes it is”, but generic and contextually irrelevant. We hypothesize that this is due to the HED model not being optimized to encode the sequence of negotiation strategies and dialogue acts.  We believe that this is the reason for a high natural score for HED. From manual inspection, we see that HED is not able to produce very persuasive responses and relies on generic responses. For example HED produces responses like : “I can do that.”, “It is new.”, “Thank you.”, “That works for me”, etc. The example in the appendix shows cases where HED uses generic statements. We have added this note in the results section of the final version.

---

### Decision · Program_Chairs · 2021-01-07
**Final Decision**

**Decision:**

Accept (Poster)

**Comment:**

In the context of constructing negotiation dialogue strategies/policies, the authors explore the use of graph attention networks (GATs) for determining the sequence of negotiation dialogue acts -- specifically leading to a (1) hierarchical dialogue encoder via pooled BERT + GRU encoding -> (2) GAT over dialogue strategies/acts (many technical details around graph usage) -> (3) GRU decoder. While a relatively straightforward replacement relative to similar architectures with other 'structural' encoders, they provide a sound end-to-end training strategy that is shown to perform well on the buyer-seller negotiation task via CraigslistBargain dataset where they demonstrate SoTA performance.

== Pros ==
+ Studying the pragmatics component of negotiation dialogue strategies has received recent interest and this seems a good milepost that demonstrates mainstream methodological approaches for this task (i.e., this is a good baseline for future innovations)
+ The paper is well-written in that it is easy to understand intuitively while having sufficient detail to understand the details.
+ The empirical results appear promising and meet the standard within this sub-community -- showing improvements with automatic and human evaluation.

== Cons ==
- This builds on existing datasets, which are known to have undesirable properties (e.g., automatic evaluation, small number of dialogue datasets, use of explicit dialogues acts, etc.) While it still meets the standards of this sub-community, it still isn't a completely convincing task.
- While the use of GATs is novel in this setting and they get it to work within the overall architecture, this is something that many people are likely trying at this time -- so there isn't an exciting 'disruptive' step here.
- The empirical results, while satisfactory from a quantitative perspective, even in reading the Appendices, it isn't clear that these are significantly better from a planning perspective or if it is just 'pattern recognition' gains.

Evaluating along the requested dimensions:
- Quality: The underlying method is fairly straightforward and the authors incorporate up-to-date GAT-related methods to get this to work in this setting. The empirical results are sound if predicated on the general quality in this sub-community where you have the standard machine translation evaluation problem for meaning vs. lexical closeness. To mitigate, they use BERTScore and human evaluation -- which is at the higher end of what can be reasonably expected.
- Clarity: The paper is written clearly overall, especially if considering the appendices where there is significant detail. Related to empirical evaluation, it isn't easy to intuitively interpret the results, but this is again par for the course. Additionally, I believe the authors did a good job responding to reviewer concerns.
- Originality: While all of the reviewers agreed that the approach was novel in this setting, one of the reviewers explicitly pointed out that using GATs in negotiation dialogues isn't that exciting -- and I mostly agree. I view this as something that somebody would have done and will serve as a good baseline; although I think this sub-field is going to need more datasets to continue progressing.
- Significance: As stated above, it is a good baseline that I think many are likely thinking of (as the TOD community has been doing this for a bit now). However, it is done well.

Honestly, I agree with the reviewers that this is a somewhat borderline paper -- mostly due to it being a fairly 'obvious' idea and the nature of the subfield making it not entirely clear if the improvements are due to knowing the target performance while training or due to the methodological advance. Personally, I am convinced, but it isn't totally clear. That being said, it is a well-written paper and I think the reviewer issues were sufficiently addressed. Thus, I would prefer to see it accepted as I think it will be a strong methodological baseline for this problem (which hopefully will accumulate more convincing datasets and standard evaluation).